# Emergent disorder and mechanical memory in periodic metamaterials

Chaviva Sirote-Katz [1,8], Dor Shohat [2,8], Carl Merrigan[3], Yoav Lahini [2,4], Cristiano Nisoli [5] & Yair Shokef [3,4,6,7] ✉

Ordered mechanical systems typically have one or only a few stable rest configurations, and hence are not considered useful for encoding memory. Multistable and history-dependent responses usually emerge from quenched disorder, for example in amorphous solids or crumpled sheets. In contrast, due to geometric frustration, periodic magnetic systems can create their own disorder and espouse an extensive manifold of quasi-degenerate configurations. Inspired by the topological structure of frustrated artificial spin ices, we introduce an approach to design ordered, periodic mechanical metamaterials that exhibit an extensive set of spatially disordered states. While our design exploits the correspondence between frustration in magnetism and incompatibility in meta-mechanics, our mechanical systems encompass continuous degrees of freedom, and thus generalize their magnetic counterparts. We show how such systems exhibit non-Abelian and history-dependent responses, as their state can depend on the order in which external manipulations were applied. We demonstrate how this richness of the dynamics enables to recognize, from a static measurement of the final state, the sequence of operations that an extended system underwent. Thus, multistability and potential to perform computation emerge from geometric frustration in ordered mechanical lattices that create their own disorder.

When interactions between different components of a complex system cannot simultaneously be minimized[1,2], the lowest-energy state is a compromise in which some elements of the system are left "unhappy". This frustration can lead to constrained disorder, degeneracy, and multistability[3–5]. Recent research translated these concepts from magnetic spin systems to engineered soft-matter systems, such as acoustic channels[6], buckled elastic beams[7], monolayers of colloidal spheres[8,9], and mechanical origami[10–15]. Here, frustration in magnetic spin systems is related to incompatibility of soft modes in mechanical systems[16–19]. Mapping of spin systems to metamaterial architectures has provided insight into the mechanical consequences of frustration,

such as stress control[20] and domain-wall topology[21,22]. In particular, exploring irreversibility and history dependence through these analogies has opened new routes for programmable elastic responses and mechanical memory storage[23–30], and has set the grounds for further advances in mechanical computing[31,32]. However, multistability and complex memory formation are typically traits of disordered and amorphous systems[27,33,34], which are harder to predict and control. On the other hand, periodic mechanical systems are generally not multistable, as their long range elastic interactions tend to resolve frustration with long-range ordered ground states[7,35]. Here, the displacement of mechanical degrees of freedom can take intermediate values,

[1]Department of Biomedical Engineering, Tel Aviv University, Tel Aviv 69978, Israel. [2]School of Physics and Astronomy, Tel Aviv University, Tel Aviv 69978, Israel. [3]School of Mechanical Engineering, Tel Aviv University, Tel Aviv 69978, Israel. [4]Center for Physics and Chemistry of Living Systems, Tel Aviv University, Tel Aviv 69978, Israel. [5]Theoretical Division, Los Alamos National Laboratory, Los Alamos, NM 87545, USA. [6]Center for Computational Molecular and Materials Science, Tel Aviv University, Tel Aviv 69978, Israel. [7]International Institute for Sustainability with Knotted Chiral Meta Matter, Hiroshima University, Higashi-Hiroshima, Hiroshima 739-8526, Japan. [8]These authors contributed equally: Chaviva Sirote-Katz, Dor Shohat. ✉e-mail: shokef@tau.ac.il

leading to an ordered compromise and lifting the degeneracy associated with frustrated spin systems.

In this article, we introduce *ordered* mechanical metamaterials that exhibit a large multiplicity of *disordered* metastable states. This leads to multistability of internal degrees of freedom and to mechanical memory, as configurations can depend on their preparation history. Crucially, our approach relies on spatial separation between the frustrated motifs in the metamaterial, which we achieve via a mapping to *vertex-frustrated*[36] artificial spin ice (ASI)[37]. As a result, long-range cooperative effects are suppressed, and disorder emerges from local rules. Furthermore, the graph of transitions between the metamaterial's states[33] contains irreversible pathways, and its structure leads to non-Abelian behavior. Namely, the system's state depends not only on the external manipulations applied on the metamaterial, but also on their precise sequence. We utilize this to demonstrate how the history of operations acted on the system, and their precise sequence, may be inferred from a static measurement of the system's final state. This takes an important step towards a systematic implementation of computation in mechanical metamaterials[38].

The design of our mechanical metamaterial is inspired by the frustration-based designs of ASI[37,39,40], which lead to constrained disorder, and thus to a degenerate manifold of configurations often captured by interesting emergent descriptions[41,42]. There are strong similarities between ASI and mechanical metamaterials realized by repeated arrangements of simple units endowed with a soft mode: In ASI, nano-islands are arranged along the edges of a lattice, and their magnetization is described by binary arrows pointing toward or from the vertices of the lattice, with certain vertex configurations locally minimizing the energy. Identically, in a mechanical metamaterial, displacements point into or out of the repeating units, and the softest deformation mode of each unit is related to a certain mutual arrangement of these displacements.

The similarity extends to frustration and incompatibility: In vertex-frustrated ASI lattices, not all vertices can simultaneously be in their lowest-energy state[36,43], as demonstrated for the Shakti-lattice ASI in Fig. 1a. In such lattices, when going along a loop of vertices around any plaquette of the lattice, the spins may not be assigned in a way such that all vertices will be in their energetically-preferred configurations. Equivalently, in combinatorial mechanical metamaterials, one can arrange the units so that they cannot all simultaneously deform according to their soft mode[16,18,19]. Vertex frustration in magnetism thus corresponds to incompatibility in mechanical metamaterials. The plethora of frustrated ASI geometries and topologies[36,40,44–46] inspires novel metamaterial designs. We show that in the mechanical system such ASI-inspired design can suppress long-range ordered ground states, and lead to emergent disorder and complex memory formation.

## The Chaco mechanical metamaterial

In the Shakti ASI[36,41,44,47,48], vertices have coordination numbers (or number of impinging arrows) $z = 2, 3, 4$. Vertices where $z = 3$ or $z = 4$ spins meet have two distinct energy-minimizing configurations [Fig. 1b], and $z = 2$ vertices prefer to have their spins aligned. Because of the vertex-frustrated lattice geometry, it is impossible to set all vertices to these lowest-energy configurations [see rectangular loop (yellow) in Fig. 1a], without resulting in a conflict (red arrows). Excitations of the $z = 3$ vertices cost less energy than excitations of the $z = 4$ vertices. Consequently, the extensively-degenerate, disordered ground state of the Shakti ASI is achieved by any distribution of $z = 3$ excitations such that each excitation resolves the frustration for two minimal loops of spins. The remaining $z = 3$ vertices and all the $z = 4$ vertices adopt minimum-energy configurations.

In Fig. 1c we introduce what we shall refer to as the Chaco-lattice mechanical metamaterial, whose incompatibility corresponds to the frustration of the Shakti ASI. The $z = 3$ and $z = 4$ vertices of the magnetic Shakti [Fig. 1b] correspond to the triangular and square units,

respectively, comprising the mechanical Chaco [Fig. 1d]. The arrows describing the magnetic moments in the lowest-energy states of the former correspond to the softest, or lowest-energy deformations of the latter. We model the mechanical Chaco as a network of linear springs. Each edge of the squares and triangles consists of two springs with stiffness $k_1$ and rest length $\ell$. These are connected by internal coupling springs of stiffness $k_2$.

In ASI, nanoislands are naturally magnetized. In metamaterials, one can induce an equivalent spontaneous displacement of all edges by prestressing the system[22]. Namely, pinning an edge to a distance $2\alpha\ell$, with $\alpha < 1$, imposes a compression, which causes it to buckle by an amount $\delta = \ell\sqrt{1 - \alpha^2}$. This results in a contraction by a factor $\alpha^2$ of the two-dimensional lattice. In the prestressed Chaco lattice, the units may not all simultaneously adopt their zero-energy deformations, due to the inherent incompatibility. Nevertheless, in the weak coupling limit, $k_2 \ll k_1$, we expect the free nodes to behave in an approximately binary way[22], adopting displacements $\vec{s}_i \approx \pm \delta \hat{z}_i$ from their rest position, where $\hat{z}_i$ is the local direction perpendicular to the edge. In this limit, the $k_1$ bonds are approximately relaxed and most of the strain is concentrated on the $k_2$ bonds. As we discuss below, the overall compression applied to the lattice breaks the symmetry between stretching and compressing, such that compressing a $k_2$ bond costs more energy than stretching it.

Experimentally, we create the elastic network shown in Fig. 1e, top. Similarly to the spring model, a thin enough elastic beam of length $2\ell$ compressed by a factor $\alpha$ tends to buckle via its first deformation mode. We prestress the rubber network by constraining the corners of all square and triangular units to an array of metal pins connected to a rigid substrate, and spaced such that the network is uniformly compressed by factor $\alpha$ [Fig. 1e, bottom]. The geometry of the units leads to mechanical soft modes [Fig. 1f], which reproduce those of the theoretical springs model [Fig. 1d]. Here, the slender beams in the triangles are coupled via their bending around a vertex. The coupling $k_2/k_1$ is related to the beam thickness near vertices (See Supplementary Information for experimental details).

## Theoretical equilibrium states

The overall compression of the springs lattice breaks the $Z_2$ spin-reversal symmetry between nearest-neighbor degrees of freedom. The mechanical degrees of freedom behave in a spin-like binary way for $k_2 \ll k_1$, whereas elastic, continuous deformations are expected otherwise. Consider the interaction energies when ideal displacements $\pm \delta$ are imposed on pairs of neighboring edges: One in and one out displacement allow the internal $k_2$ bonds to rotate and remain in their relaxed lengths $l_0^\square = \sqrt{2}\ell$ and $l_0^\Delta = \ell$. Two out displacements result in a stretched internal $k_2$ bond, with lengths $l_+^\square$ and $l_+^\Delta$, and energies $E_+^\square$ and $E_+^\Delta$. Two in displacements result in a compressed internal $k_2$ bond, with lengths $l_-^\square$ and $l_-^\Delta$, and energies $E_-^\square$ and $E_-^\Delta$. Due to the overall compression, stretching the internal bonds costs less energy than compressing them further, as shown in Fig. 2. An important consequence of this breaking of the $Z_2$ symmetry is that excited triangular and square mechanical units have distinct energy hierarchies than the $z = 3$ and $z = 4$ Shakti ASI vertices[36]. In the ideal binary limit, $k_2 \ll k_1$, the energy of a mechanical unit follows from the number of stretched and compressed bonds. There are six energy levels for the square units with energies: $0, 2E_+^\square, E_+^\square + E_-^\square, 2E_-^\square, 4E_+^\square$, and $4E_-^\square$. Triangles have five possible energies, $0, E_+^\Delta, E_-^\Delta, 2E_+^\Delta$, and $2E_-^\Delta$.

Similarly to the magnetic Shakti, the most energy-efficient way to resolve the frustration in the mechanical Chaco is to localize stress on half of the triangles. This leaves the remaining triangles and all squares close to their zero-energy states. However, the continuous nature of the mechanical degrees of freedom allows minimizing the energy by long-range pairing of the excited triangles. Namely, within two adjacent excited triangles, the $k_2$ bonds do not have to take the ideal values $l_+^\Delta$ and $l_-^\Delta$ mentioned above. Instead, due to the nonlinear dependence

of energy on displacement, there is an intermediate balance between their deformations which minimizes the energy. In a perfect realization, this leads to an ordered ground state with defect pairing [Fig. 3a], which has a four-fold degeneracy, since stresses may be placed on either vertical or horizontal pairs of triangles and since spin reversal leaves the energy unchanged. For sufficiently large $k_2/k_1$, the ground state is the only mechanically-stable configuration. However, for small

$k_2/k_1$, we also find an extensive manifold of metastable states. Namely, any configuration with all squares in one of their individual zero-energy states becomes metastable. This holds for configurations with stresses localized to exactly half of the triangles, equivalent to any ground state of the Shakti [Fig. 3b]. It also holds for configurations where more than half of the triangles are excited [Fig. 3c]. Although these states have higher energy than the ground state, the excess

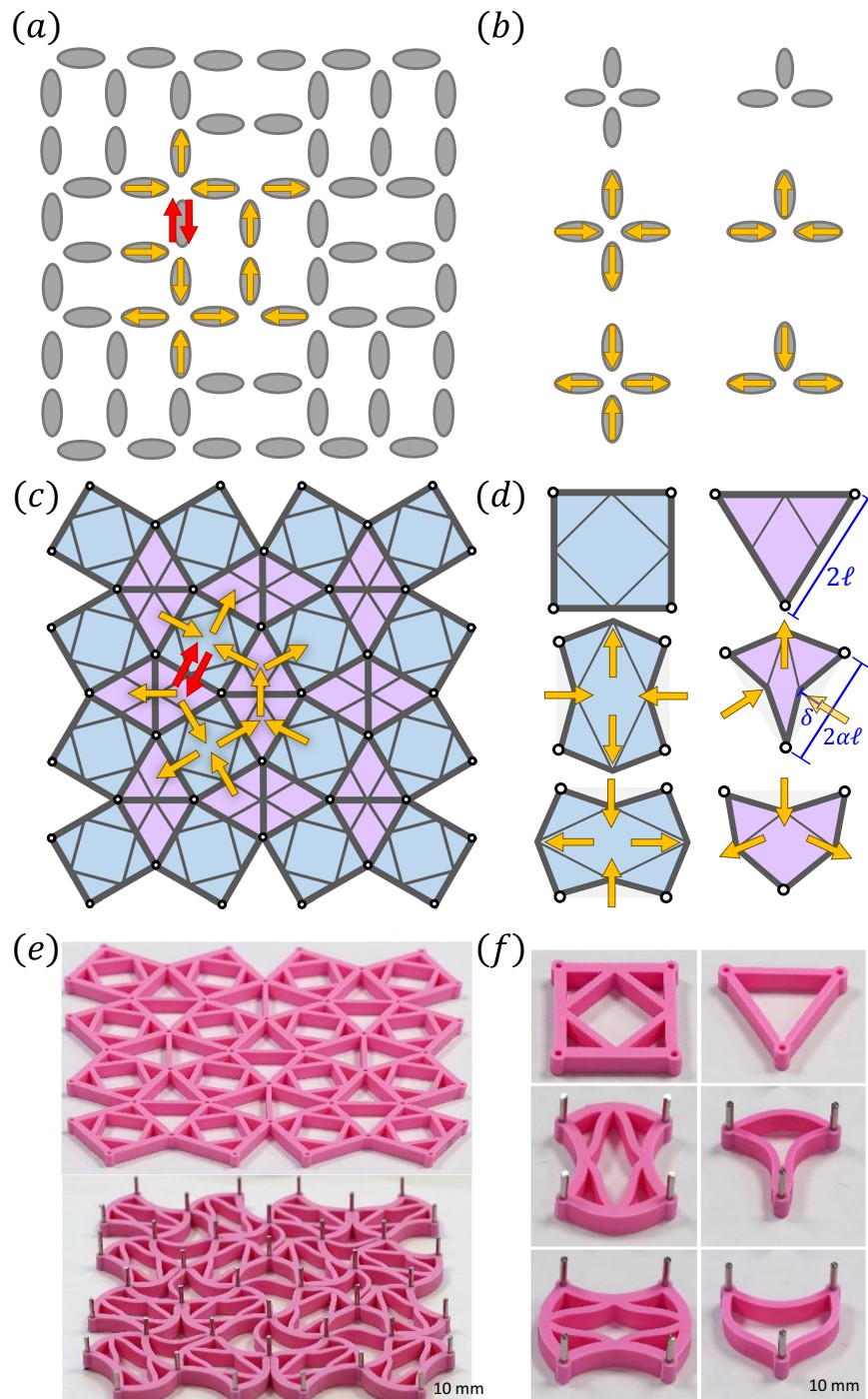

**Fig. 1 | Design principle. a** The Shakti lattice artificial spin ice (ASI) exhibits vertex frustration: A conflict (red arrows) results when trying to place all vertices around a rectangular plaquette in their lowest-energy states (as individually shown in **b**). **c** The Chaco lattice mechanical metamaterial inherits this vertex frustration of the Shakti ASI: The five units surrounding any corner in the metamaterial may not simultaneously deform according to their softest mode (as individually shown in **d**). Thick lines indicate $k_1$ springs forming the edges of the square and triangular units.

Thin lines indicate $k_2$ springs responsible for the interactions between neighboring edges within each unit. Spontaneous edge displacements are generated by pinning the corners of all units (white dots) to distances smaller than the relaxed edge length. **e** Experimental realization of the Chaco metamaterial, and (**f**) the softest deformations of its units. In the triangular units, the different stiffness at the corners connecting adjacent edges give rise to the softest modes obtained by the $k_2$ springs in the theoretical model.

stress in the triangles does not suffice to overcome the energy barrier for flipping the squares. Thus, since thermal fluctuations are negligible, if the system is in such a disordered metastable state, it will remain there and will not relax to the ordered ground state. See Supplementary Information for computational details.

Here we theoretically considered a lattice of harmonic springs, that despite their symmetric mechanical response about the relaxed state of each spring, the overall pre-compression of the lattice leads to asymmetry around its deformed state. For the experimental realization using slender elastic beams, there is an additional inherent asymmetry of each beam to buckling under compression vs. stretching under extension.

## Memory and irreversible dynamics

We distinguish between two types of degrees of freedom within the Chaco metamaterial – the square units, and the central edges between pairs of back-to-back triangles. For small $k_2/k_1$, we describe both of them as approximately binary variables, stuck in one of their stable states. Namely, squares adopt one of their soft configurations, and each central edge buckles to one of two directions. The latter degrees of freedom constitute *hysterons*[24,49,50], bistable elements whose state is history dependent. Hence, they are a natural representation for

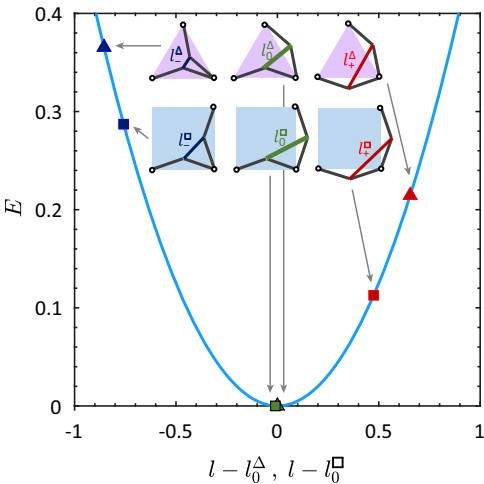

**Fig. 2 | $Z_2$ symmetry breaking.** Energetics of the ideal nearest-neighbor interactions for the square and triangular mechanical units. Ideal displacements allow the $k_1$ springs to be relaxed and place all energy on the internal $k_2$ springs. Spin inversion symmetry is broken in the mechanical system: extension costs less energy than compression.

memory in the metamaterial. As described above, both states of the squares are always stable. In contrast, the stability of the central edge between two triangles depends on the configuration of the four squares surrounding it. We label the $2^4 = 16$ possible states of these squares by $2 \times 2$ matrices with binary entries, where 0 indicates a displacement into the double triangle and 1 out of it. By symmetry, these 16 states reduce to the seven distinct configurations shown in Fig. 4a.

We now fix all squares to their zero-energy states, and compute the minimal-energy states of the central edge as function of $k_2/k_1$, for each configuration. The horizontal displacement of the central point is negligible, thus we focus on its vertical displacement $y$. We find that it may exhibit monostable, metastable or bistable behavior for the different configurations and for different values of $k_2/k_1$, as shown in Fig. 4b, c: Due to their vertical asymmetry, configurations $i - iii$ are typically monostable, with $y > 0$: configuration $i$ allows both triangles to adopt zero-energy states, such that $y = \delta$ for any $k_2/k_1$; configurations $ii, iii$ show a reduced amplitude $0 < y < \delta$, which decreases as $k_2/k_1$ increases. For very small values of $k_2/k_1$, metastable solutions with $y < 0$ appear for configurations $i - iii$. Configurations $iv - vii$ are vertically symmetric. As a result, the central edge is bistable for small $k_2/k_1$, while for larger $k_2/k_1$, the central point adopts the value $y = 0$. From here on, we focus on $0.022 \lesssim k_2/k_1 \lesssim 0.057$, where configurations $i - iii$ are strictly monostable and configurations $iv - vii$ are bistable, as shown in Fig. 4c, middle panel.

Our experimental design is similarly aimed to exhibit such behavior for all seven configurations. However, as opposed to the theoretical spring model, in the experimental system, the squares apply torque to the edges of the central beam. In configuration $vi$ the torques are not symmetric under up-down reflection, and consequentially they excite the second bending mode of the central beam, as shown in Fig. 4a. Hence, this configuration is bistable only in the theoretical springs model, while it is monostable in the experimental network of elastic beams.

Since the configuration of the squares governs the stability of the central edge, we can precisely control the states of the triangles by manipulating the squares around them. Interestingly, the final state of the central edge depends not only on the configuration of squares, but on the exact order in which they were flipped, and consequently transition pathways are sequence dependent; The transitions between states are best understood using the directed graph[24,33,50] shown in Fig. 5a, where we separately mark the two possible states of the central edge for the six bistable configurations, resulting in a total of 22 states. All transitions between states with the same direction of the central edge are reversible (gray). An irreversible transition (peach) occurs whenever a bistable configuration is connected to a monostable configuration with the opposite direction of the central edge.

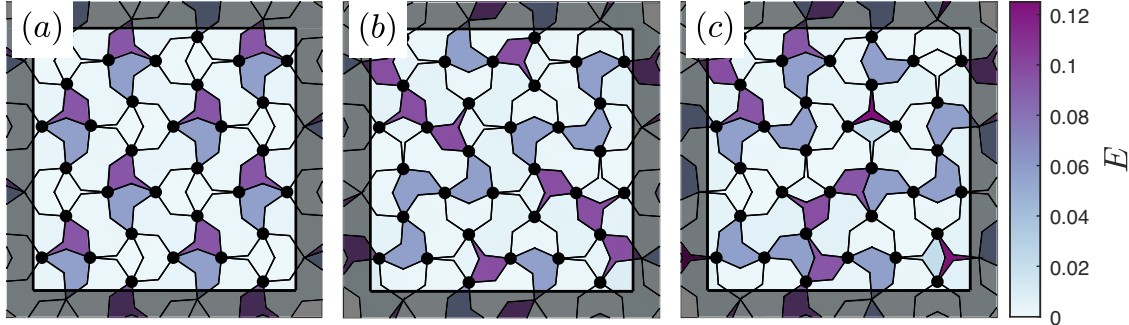

**Fig. 3 | Multiple stable configurations.** Square and triangle unit energies for sample configurations with increasing energy in a 4 × 4 Chaco lattice with periodic boundary conditions, all for $k_2/k_1 = 0.033$ and $\alpha = 0.9$: **a** Ordered ground state with energy localized on vertical pairs of triangles. **b** Complex stress distribution of a metastable state derived from a magnetic Shakti ground state, in which each stressed triangle is adjacent to a relaxed triangle. **c** Metastable configuration with all squares in relaxed random orientations and stress localized to more than half of the triangles.

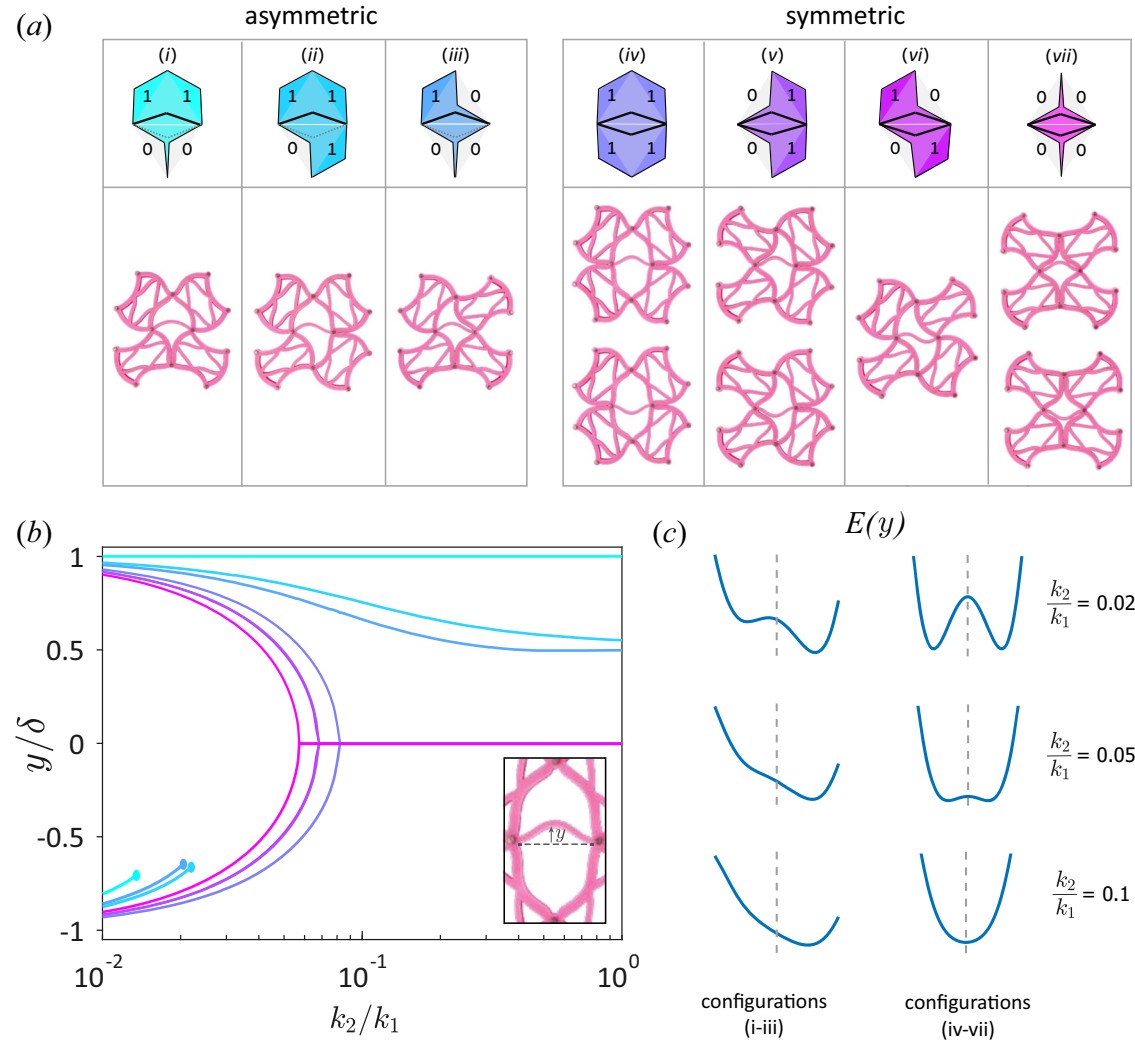

**Fig. 4 | Stability of double triangles. a** The seven distinct configurations of a pair of back-to-back triangles and their surrounding four squares within the Chaco lattice, with 1 (0) indicating displacement out (in) of the triangle. We divide them by the symmetry of forces applied to the central beam; Configurations *i-iii* are monostable. Configurations *iv,v,vii* are bistable. Configuration *vi* is bistable in the theoretical springs model, while experimentally its central beam exhibits a single stable state with a second buckling mode. **b** Stable vertical positions of the central point between the two triangles as a function of $k_2/k_1$ for $\alpha = 0.9$ and assuming that the four external edges are fixed with ideal displacements $\vec{s}_i = \pm \delta \hat{z}_i$. Inset indicates the vertical displacement $y$. **c** Potential energy $E(y)$ as function of central beam displacement for configurations *ii* (left) and *vi* (right) representing the two classes. At high $k_2/k_1$ the symmetric configurations become monostable (bottom right). At low $k_2/k_1$ the asymmetric configurations develop metastability (top left). In between is our operational regime where all configurations follow the behavior shown in (**a**).

We propose a protocol, shown in simulations and experiments in Fig. 5b, c, respectively, that demonstrates the aforementioned control: We start from the bistable configuration $\begin{pmatrix} 0 & 1 \\ 0 & 1 \end{pmatrix}$, with the central edge pointing upwards (top panels). Next, we flip the lower-left square, thus switching to the monostable state $\begin{pmatrix} 0 & 1 \\ 1 & 1 \end{pmatrix}$. This indirectly forces the central edge to flip downwards (middle panels). Finally, we flip the lower-left square back to its initial state, which brings us back to the original bistable configuration $\begin{pmatrix} 0 & 1 \\ 0 & 1 \end{pmatrix}$. Interestingly, the central edge does not revert back, but remains down (bottom panels). After this irreversible path, if we continue flipping the lower-left square, the system reversibly switches back and forth between the latter two states.

## Non-Abelian response

The subsystem of 2 × 2 squares analyzed above suffices to give rise to rich dynamics and pathways within the Chaco metamaterial. The generic motif underlying this richness, we find, is path dependency. The state of the Chaco metamaterial is sensitive to the ordering of external operations, rendering it non-Abelian. Recently, such non-Abelian responses were used to realize both finite state machines[51] and set-reset latches[26], which are elementary components of computation.

To demonstrate this here, we start from the state $\begin{pmatrix} 0 & 1 \\ 1 & 0 \end{pmatrix}$, and consider the response to flips of the surrounding squares $A, B, C, D$, as shown in Fig. 6. Upon flipping squares $A$ and $B$, the final state depends on the flipping sequence. In other words, the operators $A$ and $B$ of flipping these two squares do not commute, $AB \neq BA$.

## Sequence recognition

Next, we consider an extended Chaco lattice with 4 × 4 squares. Here, the dynamics of each double triangle is governed by the transition graph shown in Fig. 5a. We can thus understand the dynamics of a large system, based on the local rules described above. Using four different inputs, we now show that the generic non-Abelian response presented

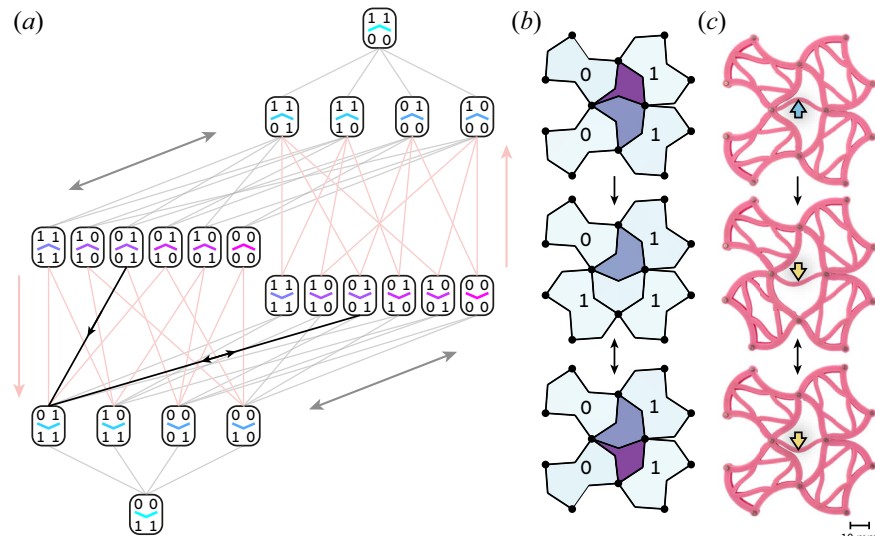

**Fig. 5 | Local memory and irreversibility. a** Transition graph between the possible states. Bistable configurations are shown twice to include the possible deflections of the central edge. Reversible transitions are plotted in grey. Irreversible transitions, which flip the central edge, are plotted in peach. **b, c** Irreversible control sequence for the internal bistable edge in simulations (**b**) and experiment (**c**). The starting configuration (top) is bistable with the internal edge up (cyan arrow).

Flipping the lower-left square (middle) creates a monostable configuration, so the internal edge flips down (yellow arrow). The lower-left square is returned to its original bistable state (bottom), but the internal edge remains pointing down. The pathway corresponding to panels (**b**, **c**) is highlighted in black on the transition graph (**a**).

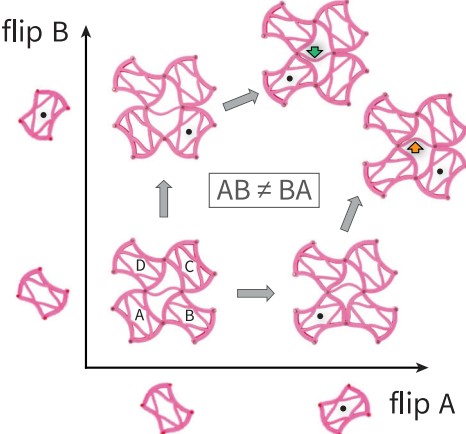

**Fig. 6 | Non-Abelian response.** Starting from state $\begin{pmatrix} 0 & 1 \\ 1 & 0 \end{pmatrix}$, consider flipping the bottom squares marked $A$ and $B$ as two operations acting on the system. The response of the Chaco metamaterial to these operations depends on the order in which they are applied, as indicated by the displacement of the central edge. Black dots mark the square that has just flipped.

in Fig. 6 can be harnessed to implement sequence recognition. Namely, after a set of square flips, the state of the double triangles may allow to infer the order of these flips and to detect a chosen sequence.

Starting from the initial configuration shown in Fig. 7a, we flip the four middle squares marked $A, B, C, D$. After flipping all four squares, the double triangles shared by each two squares encode their flipping sequence. In other words, the final state of the four relevant double triangles (marked by arrows), encodes the partial ordering of the flips of each pair of adjacent squares. Since this ordering is partial, it is insensitive to switching the order of non-neighboring flips (for example $A$ and $C$, or $B$ and $D$). Therefore, we can discern between $2^4 = 16$ out of the total $4! = 24$ possible sequences.

We now consider a target configuration, for instance the one shown in Fig. 7b, and denote it $\sigma^t$, the state vector of all double triangles. We ask which flip sequence results in this configuration. The

partial ordering between neighboring flips can be translated to a series of inequalities. Solving these inequalities allows us to analytically obtain the desired sequence, in this case $BCDA$. We can now formulate a metric $d = \sum_i (\sigma_i - \sigma_i^t)^2$ which captures the distance of a given state from the target configuration. Following $d$ along any flip sequence reveals that indeed, only the theoretically predicted sequence $BCDA$ leads to the target, as shown in Fig. 7c. All other sequences end in final configurations, which differ from the target configuration. This metric can implement sequence recognition. By following $d$ one can infer whether the sequence matched the target ($d = 0$) or was incorrect ($d > 0$) as shown in Fig. 8 and in the Supplementary Movie 1, for the correct sequence and for an example of an incorrect sequence. The color coded arrows in cyan and yellow represent the initial beam configuration and beams that have flipped, respectively.

## Discussion

The mechanism underlying the non-Abelian response sheds light on the generic emergence of sequence-dependent responses in frustrated mechanical systems[25,26,52–54]. The order in which bistable degrees of freedom are manipulated follows a pathway of states, which favors one of the possible final states. In essence, the sequence directs the otherwise spontaneous symmetry breaking of the system at its final state.

The frustrated geometry we have presented can be realized in a wide range of soft mechanical systems. Internal stresses and buckling can be implemented by thermally activated polymers[55,56], inflatable structures[57,58] or via deflation[26], allowing to couple global driving with local programmability. This can be used to cleverly manipulate mechanical systems between multiple functionalities[59–61], as different states may exhibit different global responses. This, we envision, may allow adaptable, history dependent mechanical responses via *in materia* information processing[31,32].

We have focused theoretically and experimentally on the weak interaction limit (measured by the coupling parameter $k_2/k_1$). Strong interactions between hysterons lead to long-range order and a distinct ordered ground state[7], see Supplementary Information. However, intermediate interaction strengths may give rise to even richer dynamics[24,27]. For example, they can lead to long transients, or

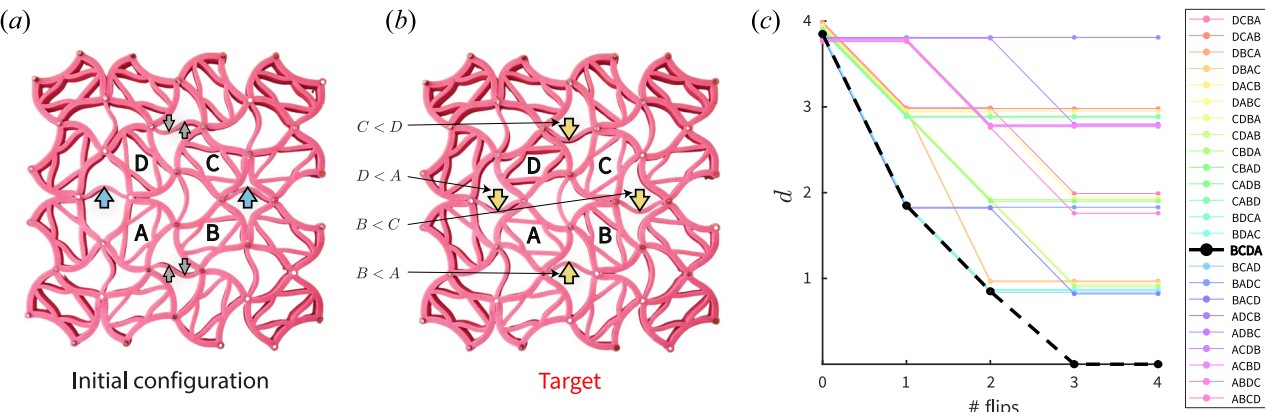

**Fig. 7 | Sequence recognition. a** Initial configuration for the sequence recognition protocol, where the squares marked $A − D$ are flipped. **b** Target configuration after all squares have been flipped. The state of the double triangles, denoted $\sigma^0$ translates to a set of inequalities for the flipping order; (**c**) distance from the target $d = \sum_i (\sigma_i − \sigma_i^t)^2$ along any flip sequence. The sequence $BCDA$ which satisfies all inequalities is the only one reaching the target, and the distance metric allows sequence recognition.

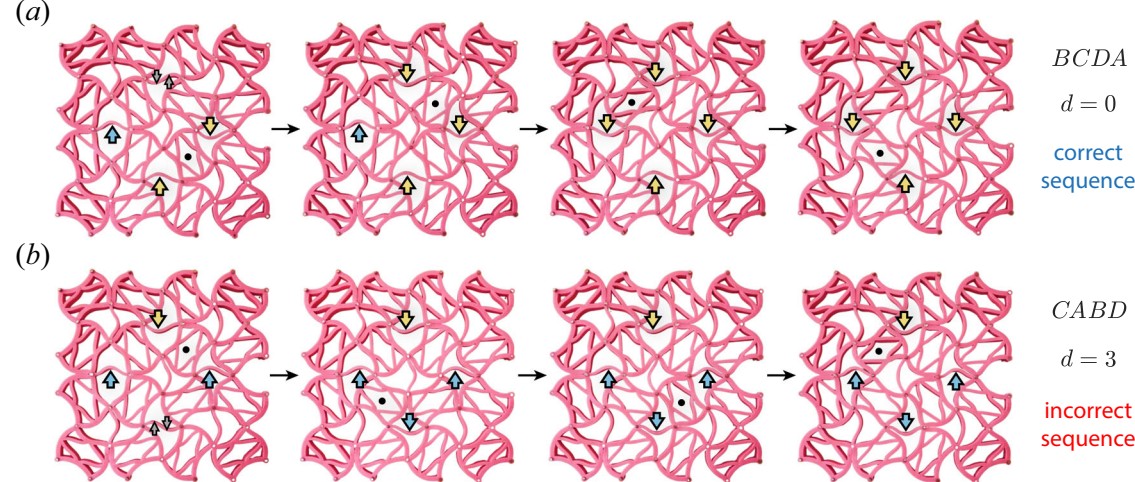

**Fig. 8 | State evolution during flip sequences.** Starting from the initial condition of Fig. 7a, we track the state of the system along two sequences. Black dots mark the square that was flipped in each stage. The correct sequence $BCDA$ (**a**) leads to the target state, or $d = 0$; in contrast, an incorrect sequence $CABD$ (**b**) results in the wrong state, measured by the metric $d = 3$.

complex cycles with longer periods[62–64], which can realize an array of computational tasks such as counting[65].

Following the vertex-frustration analogy back to the magnetic realm, non-Abelian protocols may increase the memory capacity in storage devices based on frustrated geometries. Due to the sequence sensitivity, $n$ binary operations (bit flips) can access more than $2^n$ distinct states, as the system's history may influence additional features that are beyond the states of the manipulated degrees of freedom. Thus the manifold of reachable states may be enriched by frustrated interactions.

Finally, the Chaco mechanical metamaterial inherits from the Shakti ASI a topological structure: the possible allocation of its incompatibilities can be mapped both in the free fermion point of a six vertex model[47] and thus into a dimer cover model[48]. These height models can be thought of as instances of so-called classical topological order[66] for which boundary conditions strongly constrain the manifold in the bulk, and in the case of the Chaco metamaterial, that would imply an encoding via the boundaries. Moreover, using the same dimer mapping and the design strategy by triangle rotation pioneered in ref. 18 for the kagome lattice, an extensive number of Chaco-based metamaterials can be produced.

## Data availability

The data supporting the findings of this study are included within the paper and its Supplementary Information.

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

## Acknowledgements

We thank Ben Pisanty, Eial Teomy, Erdal Oğuz, Izhar Neder, Muhittin Mungan, Ofer Shochet, Priyanka, Roni Ilan, and Yael Roichman for fruitful discussions. This research was supported in part by the Israel Science Foundation Grants No. 1899/20 (Y.S.) and 2117/22 (Y.L.). Y.S. and C.M. thank the Center for Nonlinear Studies at Los Alamos National Laboratory for its hospitality. The work of C.N. was carried out under the auspices of the U.S. DOE through the Los Alamos National Laboratory, operated by Triad National Security, LLC (Contract No.229892333218NCA000001) and funded by a grant from the DOE-LDRD office. C.S-K. and D.S. acknowledge support from the Clore Israel Foundation.

## Author contributions

C.N. and Y.S. suggested the Chaco metamaterial. C.M., C.N. and Y.S. theoretically analyzed the equilibrium states. C.M. designed and performed the numerical simulations. C.S-K., D.S., Y.L. and Y.S. analyzed the transition graph and identified the dynamical phenomena on it. C.S-K., D.S. and Y.L. designed and performed the experiments. All authors wrote the paper.

## Competing interests

The authors declare no competing interests.
