## [Peer Review File · Nature Communications]

Reviewers' Comments:

Reviewer #1:

Remarks to the Author:

This manuscript presents the study on the disordered frustrated states of a periodic lattice, and shows that the non-Abelian response of frustrated lattices enables memory and computations. This work is interesting and relates to the emerging topic of morphological computing. The results are clearly presented and supported. However, to reach a broader range of audience that it currently delivers, some details on the mechanical setup needs to be further clarified. I have the following comments for the authors to consider:

1. The current literature review is rather limited to the physics side of research on frustrated metamaterials. Some advances from the mechanical engineering side should be included. For example, topics related to buckled/multi-stable lattices, origami, and kirigami. In fact, in many multistable metamaterials, there exist local symmetry breaking that transforms the initially periodic system into aperiodic geometry (see *Adv. Mater.* 2022, 34, 2107998).
2. The authors simplify the buckling of beams into discrete kinks with 0 rotational stiffness. This is a questionable simplification. Bending energy of the beams (including regions near the lattice points) are completely ignored which may cause the calculated energy inaccurate. Since the authors claim that their research is backed by experiments, it may be necessary to include some quantitative experimental measurements to compare the energy computed by the simplified theoretical model and the actual physical model. This will also help to build a physical interpretation of the idealized stiffness parameters k_1 and k_2 .
3. Page 6: It sounds incorrect to say "overall compression applied to the lattice breaks the symmetry between stretching and compressing." The reason for the breaking of symmetry is related to the nonlinearity of the equilibrium equation of slender beams rather than overall compression. For very stiff materials, even very large compression may not generate buckling.
4. When studying the frustrated states, the authors essentially applied force to each lattice point by pinning it to a specific location. Please comment on how this boundary condition may be realized practically when the system is super large or unmanned.
5. Did the authors encounter any self-locking in large systems?

Reviewer #2:

Remarks to the Author:

In the submission, "Emergent Disorder and Mechanical Memory in Periodic Metamaterials", the authors invent a novel design for a two-dimensional mechanical metamaterial, in which an ordered, periodic lattice gives way to a plethora of spatially-disordered locally-stable states. They are inspired by the Shakti artificial spin ice in their choice of lattice. They construct a physical model of their system using silicone rubber in a 3D printed mold, and they analyze a simplified model of the system (that neglects one of the three "hinge" energies). Perhaps the most exciting feature they demonstrate is that of "sequence recognition", where a set of four bistable beams can recognize up to 16 of the possible 24 sequences of flipping the four square units around them, where each of the 4 square units is flipped exactly once.

The work appears to be thorough and carefully performed, and the results support the paper's conclusions. The experimental system is original and creative. It is novel because it possesses disordered states, despite being ordered in its (uncompressed) lattice structure. And, it displays an interesting history dependence that allows to decipher a set of sequences of inputs.

The submission also complements a growing literature on memory and computing in mechanical systems and in disordered materials. This manuscript provides an appealing architecture that can provide a platform for designing other responses and functionalities in the future. In particular, whereas other recent works have used buckled beams for mechanical computing, this lattice offers a new way for beams to couple to nearby beams without direct steric contact. For these reasons, I think this contribution is suitable for publication in *Nature Communications*, after addressing the comments below.

Specific comments:

-The abstract states "our mechanical systems encompass continuous degrees of freedom, and are hence richer than their magnetic counterparts." In what way is this richer? The stable states are discrete in the metamaterial, even if the deformations between them are continuous. I didn't see where the continuous nature of the system adds any novel features. And although the spring constants (k_1 , k_2) can be continuously tuned, so can the interactions in a spin glass (the J_{ij}).

-The final sentence of the first paragraph is unclear to me: "Here, the displacement of mechanical degrees of freedom can take intermediate values, leading to an ordered compromise and lifting the degeneracy associated with frustrated spin systems". What are the intermediate values? What is the ordered compromise?

-What is the origin of the term "Chaco"? Is it inspired by a technical term, or a name? Depending on the answer, it could be helpful to point out the origin or inspiration.

-Up to figure 6, it appears that yellow arrows denote a "down" state and blue arrows denote an "up" state. But, on figure 7 and after, yellow arrows indicate a spin that matches the target state, and blue arrows are those that differ from the target state. Gray arrows always seem to mean the "mixed" state. I found this color-coding confusing. (The supplementary movie also uses the same color coding as figure 7 and on.) Perhaps it could be either explained more clearly, or different pairs of colors could be used in these two portions of the paper? One option would be to use red and blue for to denote "correct" and "incorrect" spins, as this is the color choice used on the words on the right of Fig. 8.

-In the penultimate paragraph, the authors point out that n degrees of freedom may encode more than 2^n states. I think this phrasing is potentially misleading. If I understand correctly, here the degrees of freedom are the "square units", but the counting of states includes the triangular units as well (namely, the beams at their bases). So, one is manipulating a smaller set of binary elements (n of them), and looking at the information storage of a larger number of binary elements. I am worried that the wording in the paragraph confuses this matter (i.e., even if an element of a system is not externally controlled, it is still commonly regarded as a degree of freedom). I think it can be edited to highlight what is unique about the system, without causing this potential confusion.

Response to the Report of Reviewer #1:

We thank the Reviewer for the positive assessment, the careful reading and the constructive critique that has helped us to improve the manuscript. Enclosed please find our revised manuscript, in which we have highlighted the revision in blue color, and in what follows please find the reviewer's report followed by our detailed response to each of the comments made in it.

Reviewer:

This manuscript presents the study on the disordered frustrated states of a periodic lattice, and shows that the non-Abelian response of frustrated lattices enables memory and computations. This work is interesting and relates to the emerging topic of morphological computing. The results are clearly presented and supported. However, to reach a broader range of audience that it currently delivers, some details on the mechanical setup needs to be further clarified. I have the following comments for the authors to consider:

Response:

We thank the Reviewer for the positive assessment of our work. Below we address the Reviewer's comments.

Reviewer:

1. The current literature review is rather limited to the physics side of research on frustrated metamaterials. Some advances from the mechanical engineering side should be included. For example, topics related to buckled/multi-stable lattices, origami, and kirigami. In fact, in many multistable metamaterials, there exist local symmetry breaking that transforms the initially periodic system into aperiodic geometry (see Adv. Mater. 2022, 34, 2107998).

Response:

We agree with the Reviewer and we have added to the introduction further connections to related works in mechanics. Specifically, Silverberg 2014, Shan 2015, Findeisen 2017, Dieleman 2019, Liu 2019, and Liu 2022.

Reviewer:

2. The authors simplify the buckling of beams into discrete kinks with 0 rotational stiffness. This is a questionable simplification. Bending energy of the beams (including regions near the lattice points) are completely ignored which may cause the calculated energy inaccurate. Since the authors claim that their research is backed by experiments, it may be necessary to include some quantitative experimental measurements to compare the energy computed by the simplified theoretical model and the actual physical model. This will also help to build a physical interpretation of the idealized stiffness parameters k_1 and k_2 .

Response:

We thank the Reviewer for suggesting to strengthen the connection between our theoretical model and the actual mechanical properties of the experimental system. We experimentally designed the metamaterial to include rotational stiffness such that two adjacent beams prefer to bend and rotate in the same direction around their common pin. In this way, there is an energetic cost for changing the angle between every pair of neighboring beams. This interaction corresponds to the internal spring of stiffness k_2 in the theoretical model. In the revised manuscript, we added an explanation of this connection between the springs model and the mechanical design at the end of Section II. Since we pin all corners to a lattice which is smaller than the rest length of the beams, all beams must already be bent. This bent configuration serves as our reference point for the elastic energy of the material, and we consider only excitations above this level. Therefore, we do not need to explicitly deal with the bending energy of the beams. We would like to clarify that the energetic calculations in the paper are restricted to the theoretical springs model, and we do not get into quantitative

analysis of the full elastic problem of the experimental metamaterial. To connect between the geometrical design parameters of the experimental system and the dimensionless interaction parameters in the theoretical model, we did consider multiple designs until we obtained a metamaterial that behaves according to behavior in the desired range of k_2/k_1 from the theoretical model.

Reviewer:

3. Page 6: It sounds incorrect to say “overall compression applied to the lattice breaks the symmetry between stretching and compressing.” The reason for the breaking of symmetry is related to the nonlinearity of the equilibrium equation of slender beams rather than overall compression. For very stiff materials, even very large compression may not generate buckling.

Response:

We would like to clarify that this sentence in the paper refers to the theoretical model of linear, Hookean springs. To make this clear we changed the title of Section III in the paper, where this is discussed to “Theoretical Equilibrium States”, and also now explicitly state that this section treats the lattice of springs. Certainly the springs considered in this section exhibit symmetry between compression and stretch when in a relaxed state. And yet, in our case, the overall compression of the lattice due to the pinning at the corners of the units, necessarily breaks this symmetry. We agree that for slender elastic beams there is an additional asymmetry coming from non-linearity, as we now detail at the end of Section III. However, the theoretical analysis included in the paper is for a network of linear springs. It would indeed be interesting to further investigate the elastic behavior of the actual geometric design of our metamaterials, in which the aforementioned asymmetry in the non-linear response of the beams would be important.

Reviewer:

4. When studying the frustrated states, the authors essentially applied force to each lattice point by pinning it to a specific location. Please comment on how this boundary condition may be realized practically when the system is super large or unmanned.

Response:

We thank the Reviewer for raising this question regarding the scalability in implementation of the metamaterial that we have introduced. In larger systems, one would need to design alternative mechanisms for inducing internal stresses, so that the slender beams would spontaneously buckle. We theoretically considered inducing prestress by connecting the corners of each square or triangular unit by harmonic springs of stiffness k_3 , which have a shorter relaxed length. The rigid pinning of these corners that we experimentally implemented is the limit where $k_3 \rightarrow \infty$. Alternatively, prestress equivalent to finite k_3 can be implemented by casting a material that would swell or contract selectively, e.g. by thermal activation of polymers, as has been done in: [Klein 2007] or alternatively in [Ziv Sharabani 2022], by inflating systems, as has been done with tunable kirigami patterns in [Jin 2020] and origami structures in [Melancon 2022], or by vacuum deflation as has been done in [Guo 2023]. We address this important point in the discussion section of the revised manuscript.

Reviewer:

5. Did the authors encounter any self-locking in large systems?

Response:

If we understand correctly, the Reviewer is asking if beams in the Chaco metamaterial can jam and thus limit the possible motion of neighboring beams due to contact between them, as often appears for instance in origami metamaterials [Fang 2016; Fang 2018]. In principle, this can change the mechanical properties of the lattice. Our system is compressed uniformly by a moderate factor of $\alpha = 0.92$, far from substantial contacts forming between beams. Because of that, we did not observe any self-locking. It would be interesting to consider larger deformations and more extreme compression factors in future works. Dynamically induced

self-locking directly ties into the previous question regarding the generation of internal stresses without resorting to pinning. The alternative pre-stressing mechanisms outlined above can allow other mechanical manipulations, like shear or overall compression that could give rise to self-locking and interesting effects. We have added a note regarding the absence of contact between beams or self-locking effects in the experimental methods section.

Response to the Report of Reviewer #2:

We thank the Reviewer for the positive assessment, the careful reading and the constructive critique that has helped us to improve the manuscript. Enclosed please find our revised manuscript, in which we have highlighted the revision in blue color, and in what follows please find the reviewer's report followed by our detailed response to each of the comments made in it.

Reviewer:

In the submission, "Emergent Disorder and Mechanical Memory in Periodic Metamaterials", the authors invent a novel design for a two-dimensional mechanical metamaterial, in which an ordered, periodic lattice gives way to a plethora of spatially-disordered locally-stable states. They are inspired by the Shakti artificial spin ice in their choice of lattice. They construct a physical model of their system using silicone rubber in a 3D printed mold, and they analyze a simplified model of the system (that neglects one of the three "hinge" energies). Perhaps the most exciting feature they demonstrate is that of "sequence recognition", where a set of four bistable beams can recognize up to 16 of the possible 24 sequences of flipping the four square units around them, where each of the 4 square units is flipped exactly once.

The work appears to be thorough and carefully performed, and the results support the paper's conclusions. The experimental system is original and creative. It is novel because it possesses disordered states, despite being ordered in its (uncompressed) lattice structure. And, it displays an interesting history dependence that allows to decipher a set of sequences of inputs.

The submission also complements a growing literature on memory and computing in mechanical systems and in disordered materials. This manuscript provides an appealing architecture that can provide a platform for designing other responses and functionalities in the future. In particular, whereas other recent works have used buckled beams for mechanical computing, this lattice offers a new way for beams to couple to nearby beams without direct steric contact. For these reasons, I think this contribution is suitable for publication in Nature Communications, after addressing the comments below.

Response:

We thank the Reviewer for the positive assessment of our results and for appreciating the collective nature of our system. Below we address the Reviewer's comments.

Reviewer:

-The abstract states “our mechanical systems encompass continuous degrees of freedom, and are hence richer than their magnetic counterparts.” In what way is this richer? The stable states are discrete in the metamaterial, even if the deformations between them are continuous. I didn't see where the continuous nature of the system adds any novel features. And although the spring constants (k_1 , k_2) can be continuously tuned, so can the interactions in a spin glass (the J_{ij}).

Response:

We thank the Reviewer for bringing up this issue. We emphasize that the important difference between continuous mechanical systems and discrete spin systems is not that parameters, like k_1 and k_2 can take continuous values, but that the state of the system, which here is the displacements, is continuous. For instance, in Fig 3 we show how pairing of excited triangles allows the element between them to find an optimal deformation, which differs from that found for isolated excited triangles, thus lowering the overall energy. Similarly, in Fig 4 we show that the beam at the center of different double triangle configurations is displaced by different magnitudes. This richness is also apparent when we examine the behavior as a function of k_2/k_1 in Fig 4. The bifurcations that destroy long range order and give rise to multistability, represent a more complicated scenario compared to a discrete magnetic system. We agree with the Reviewer that the word “richer” may not be the best term to express this, and we have modified this sentence in the abstract to state that continuous systems constitute a generalization of their discrete counterparts.

Reviewer:

-The final sentence of the first paragraph is unclear to me: “Here, the displacement of mechanical degrees of freedom can take intermediate values, leading to an ordered compromise and lifting the degeneracy associated with frustrated spin systems”. What are the intermediate values? What is the ordered compromise?

Response:

Following the discussion in our response to the previous comment, the paired defected triangles shown in Fig 3a are in configuration ν of Fig 4a. Figure 4b shows that the displacement of the central point between two such triangles is smaller than the discrete value δ . This is the intermediate value of the displacement that we refer to in the sentence that the Reviewer mentioned. And, the pairing of defected triangles appearing in Fig 3a is the ordered ground state of the mechanical system, as opposed to the disordered states shown in Fig 3b. In the magnetic shakti, due to the discrete nature of its degrees of freedom, the states corresponding to both the ordered state of Fig 3a and the disordered state of Fig 3b have degenerate energy.

Reviewer:

-What is the origin of the term “Chaco”? Is it inspired by a technical term, or a name? Depending on the answer, it could be helpful to point out the origin or inspiration.

Response:

Chaco is the name that we gave to the lattice that we introduced in this paper. It is named after Chaco Canyon in New Mexico. Located about two hours from Los Alamos, it is one of the most remarkable and mysterious sites of native archeology in North America, and dear to the last two authors. The first design that translated the Shakti artificial spin ice into a mechanical metamaterial provided by CN was called Acoma (a Pueblo between New Mexico and Arizona). YS, in Los Alamos for a sabbatical at that time, provided essential modifications and the new design was named Chaco. Sometimes it would be called “Mechanical Shakti” as a working name, but we feel that it deserves a name of its own. There are other materials named after New Mexican places, such as Santa Fe Spin Ice [Science 380.6644 (2023): 526-531] whereas a Taos Nanomagnet has just been fabricated and is currently under study. We do not think that it would be necessary to discuss in the manuscript the above origin of the name, and therefore only provide this explanation to the Reviewer here.

Reviewer:

-Up to figure 6, it appears that yellow arrows denote a “down” state and blue arrows denote an “up” state. But, on figure 7 and after, yellow arrows indicate a spin that matches the target state, and blue arrows are those that differ from the target state. Gray arrows always seem to mean the “mixed” state. I found this color-coding confusing. (The supplementary movie also uses the same color coding as figure 7 and on.) Perhaps it could be either explained more clearly, or different pairs of colors could be used in these two portions of the paper? One option would be to use red and blue for to denote “correct” and “incorrect” spins, as this is the color choice used on the words on the right of Fig. 8.

Response:

We thank the Reviewer for pointing out this point which could have been confusing for the reader. The yellow and blue arrows signify beams that have changed their state, and beams that remain in their initial state respectively. To remove ambiguity, we now explicitly explain this color-coding in section VI. We have also changed the arrow colors in Fig. 6, which was inconsistent with the color-coding.

Reviewer:

-In the penultimate paragraph, the authors point out that n degrees of freedom may encode more than 2^n states. I think this phrasing is potentially misleading. If I understand correctly, here the degrees of freedom are the “square units”, but the counting of states includes the triangular units as well (namely, the beams at their bases). So, one is manipulating a smaller set of binary elements (n of them), and looking at the information storage of a larger number of binary elements. I am worried that the wording in the paragraph confuses this matter (i.e., even if an element of a system is not externally controlled, it is still commonly regarded as a degree of freedom). I think it can be edited to highlight what is unique about the system, without causing this potential confusion.

Response:

To avoid confusion, we have rephrased that sentence. It now states that “ n binary operations (bit flips) can access more than 2^n distinct states”. This captures the “enhanced” memory capacity due to sequence dependence, without counting degrees of freedom.

Reviewers' Comments:

Reviewer #1:

Remarks to the Author:

The authors have addressed my comments.

Reviewer #2:

Remarks to the Author:

We thank the authors for their careful consideration and reply to the reviewer's comments.

One minor comment:

On page 9 in the revised text immediately before section IV., the authors write: "there is an additional inherent asymmetry of each beam to stretching vs. bending." If I understand this statement correctly, then I think it would be more clear to say "stretching vs. buckling" or perhaps even more explicitly, "buckling under compression vs. stretching under extension"

As I mentioned in the previous round of reviews, I feel the main novelty of the work is the design and demonstration of a metamaterial with significant frustrated interactions between mechanical hysteron. After the revisions, the description of the work is more clear, and I feel the manuscript is now ready for publication.